# Effects of convective ice evaporation on interannual variability of tropical tropopause layer water vapor

Hao Ye, Andrew E. Dessler, and Wandi Yu

Department of Atmospheric Sciences, Texas A&M University, College Station, USA

*Correspondence to:* Andrew E. Dessler (adessler@tamu.edu)

**Abstract.**

Water vapor interannual variability in the tropical tropopause layer (TTL) is investigated using satellite observations and model simulations. We breakdown the influences of the Brewer-Dobson circulation (BDC), the quasi-biennial oscillation (QBO), and the tropospheric temperature ($\Delta T$) on TTL water vapor as a function of latitude and longitude using a 2-dimensional multivariate linear regression. This allows us to examine the spatial distribution of the impact of each process on TTL water vapor. In agreement with expectation, we find that the impacts from the BDC and QBO act on TTL water vapor by changing TTL temperature. For $\Delta T$, we find that TTL temperatures alone cannot explain the influence. We hypothesize a moistening role for the evaporation of convective ice from increased deep convection as the troposphere warms. Tests using a chemistry-climate model, the GEOSCCM, support this hypothesis.

## 1   Introduction

Stratospheric water vapor plays an important role in both the chemistry (Dvortsov and Solomon, 2001; Stenke and Grewe, 2005) and radiative energy budget (Forster and Shine, 1999; Solomon et al., 2010; Dessler et al., 2013) of the atmosphere. Air enters the stratosphere from the tropical troposphere mainly through the tropical tropopause layer (TTL, ∼15-18 km) (Sherwood and Dessler, 2000; Fueglistaler et al., 2009), which serves as a transition region between the troposphere and stratosphere. It is generally recognized that the coldest temperatures in the TTL act like a "cold trap" that provides primary control on the amount of water vapor entering the lower stratosphere (Mote et al., 1996; Holton and Gettelman, 2001). Large interannual variations of TTL water vapor have been observed and attributed to a set of physical processes that affect water vapor by varying TTL temperatures, such as the quasi-biennial oscillation (QBO) (Geller et al., 2002; Fueglistaler and Haynes, 2005; Liang et al., 2011; Liess and Geller, 2012; Randel and Jensen, 2013; Wang et al., 2015; Tao et al., 2015) and the Brewer-Dobson circulation (BDC) (Randel et al., 2006; Calvo et al., 2010; Dessler et al., 2013, 2014; Fueglistaler et al., 2014; Gilford et al., 2016).

Another important process is the deep convection that reaches the TTL. Clouds comprised of convective ice can have important impacts on planetary energy balance (Lee et al., 2009; Zhou et al., 2014), and their evaporation can moisten the TTL (Corti et al., 2008; Wang and Dessler, 2012). The efficiency of cloud evaporation is strongly related to ambient relative humidity (Dessler and Sherwood, 2004; Wright et al., 2009) because high relative humidity inhibits evaporation. Recent aircraft

measurements (Anderson et al., 2012; Herman et al., 2017) and satellite observations (Dessler and Sherwood, 2004; Schwartz et al., 2013; Sun and Huang, 2015) confirm that the deep convection enhances lower stratospheric water vapor over the North American summer monsoon region, where relative humidity is low.

In the tropics, the influence of convection on observed water vapor amounts is less clear. It seems certain that convective ice evaporation at least occasionally moistens the stratosphere (Khaykin et al., 2009; Hassim and Lane, 2010; Carminati et al., 2014; Frey et al., 2015; Virts and Houze, 2015), but the impact of convection there is muted because the relative humidity of the TTL is high, suppressing evaporation, and only convection reaching above the cold point is likely to significantly impact the humidity of the stratosphere (Dessler et al., 2007).

Several modeling studies have addressed this by adding convection moistening into the trajectory model simulation, through a convective probability scheme (Dessler et al., 2007; Schoeberl and Dessler, 2011) or a reanalysis-based anvil ice scheme (Schoeberl et al., 2014) or an observation-based convective cloud top scheme (Ueyama et al., 2014, 2015). All of these are in agreement that the convective ice can moisten the TTL and lower stratosphere. Schoeberl et al. (2014) and Ueyama et al. (2015) estimated that convective ice evaporation increases TTL water vapor by 0.3 ppmv and 0.5 to 0.6 ppmv, respectively — about a 10-15% effect. In addition, a recent case study has shown that evaporation of convective ice could account for a significant part of the TTL water vapor response to the strong El Niño of 2015-2016 (Avery et al., 2017).

On longer time scales, the impact of ice evaporation on stratospheric water vapor could be much more important. Almost all climate models predict that the water vapor in the UTLS will increase over the next century(Gettelman et al., 2010), and a significant fraction of this increase was found to be due to the evaporation of convective ice from convection in two chemistry-climate models (Dessler et al., 2016). This gives us ample motivation to look more closely at the impact of convective ice evaporation on TTL water vapor in the observations.

The purpose of this study is to investigate in more detail the physical processes controlling the interannual variations of water vapor in the TTL, particularly the influence of evaporation of convective ice. Previous work has mostly taken a "forward model" approach — where a model (usually a dynamical model coupled to a microphysical model) driven by observations of winds, temperatures, and convection, is used to make an explicit estimate of the convective influence. Our analysis takes a different approach — we use a statistical model to decompose water vapor variability into the dominant physical processes known to drive water vapor. We do this in both observations and water vapor simulated by a trajectory model. Because the trajectory model does not include convection, differences in the results will be tied to the influence of convection. We verify the methodology by reproducing it in a chemistry-climate model with known convection.

## 2 Data and Methods

### 2.1 MLS water vapor

The observations of TTL water vapor are from the Earth Observing System (EOS) Aura Microwave Limb Sounder (MLS) (Lambert et al., 2007; Read et al., 2007). The MLS instrument has obtained continuous high-quality global observations of water vapor in the upper troposphere and stratosphere since August 2004. The data are available from https://mls.jpl.nasa.gov/.

Here we use MLS version 4.2 level 2 water vapor retrievals from August 2004 to December 2016. The daily water vapor mixing ratio measurements are binned and averaged to produce monthly data on a $4° \times 8°$ latitude and longitude grid with the quality control following the instruction in Livesey et al. (2017). We focus on the interannual anomalies of water vapor from 30°N to 30°S at 100 hPa. Throughout this paper, the interannual anomalies at each grid point are calculated by subtracting the average annual cycle at that grid point.

## 2.2 GEOSCCM

We also use simulations of TTL water vapor from the Goddard Earth Observing System Chemistry Climate Model (GEOSCCM) in this study. The state-of-the-art GEOSCCM includes the GEOS-5 atmospheric general circulation model (Molod et al., 2012) with a single-moment cloud microphysics scheme (Bacmeister et al., 2006; Barahona et al., 2014) and the StratChem stratospheric chemical mechanism (Pawson et al., 2008; Oman and Douglass, 2014). The GEOSCCM simulation provides long term simulations of temperature, water vapor, horizontal winds, diabatic heating rates, and convective ice content with a resolution of $2° \times 2.5°$ in latitude and longitude on 72 vertical model levels, up to 0.01 hPa.

In this study, we investigate water vapor simulated by the GEOSCCM in the TTL during model years corresponding to the MLS period. As these simulations are from a free-running model, climate variability in the model is not synchronous with that in the observations, so the comparisons with MLS observations are done statistically — using regression models (discussed below).

## 2.3 Trajectory model

We also produce simulations of TTL water vapor using a domain-filling forward trajectory model, which has been in previous work to reproduce water vapor, ozone, and carbon monoxide anomalies in the TTL and lower stratosphere (Schoeberl and Dessler, 2011; Schoeberl et al., 2012, 2013; Dessler et al., 2014; Wang et al., 2014).

This model uses Bowman's trajectory code (Bowman, 1993; Bowman and Carrie, 2002). The parcels are driven by 6-hourly horizontal winds and total diabatic heating rates from either reanalysis datasets or from the GEOSCCM. When comparing to MLS data, we use trajectory runs driven by two reanalysis datasets: the European Centre for Medium-Range Weather Forecasts (ECMWF) ERA-interim reanalysis (ERAi) (Dee et al., 2011) and the NASA's Modern-Era Retrospective-Analysis for Research and Applications Version-2 (MERRA-2) (Bosilovich et al., 2016). When comparing to GEOSCCM output, we drive the trajectory model with meteorological fields from the GEOSCCM.

In all simulations, 1350 parcels are initialized every day from January 2000 to December 2016 on an equal area grid from 60°N to 60°S. The parcels are released on the 370-K isentropic level, which is just above the zero net diabatic heating level over the tropics ($\sim$355-360 K) but below the cold point ($\sim$375-380 K). Each parcel travels forward following the horizontal winds and diabatic heating rate. Once a parcel has a pressure larger than 250 hPa, it is regarded as having descended back into the troposphere and is removed from the model.

Each parcel is initialized with a water vapor mixing ratio of 200 parts per million by volume (ppmv). Along the trajectory, a parcel will immediately be dehydrated to saturation once its water vapor mixing ratio exceeds a predetermined saturation

threshold, 100% in this study. The saturated water vapor mixing ratio is obtained from the thermodynamic equation with respect to ice (Murphy and Koop, 2005) based on temperatures from reanalyses or the GEOSCCM. The production of water vapor from methane oxidation is also included in these trajectory model runs but it has very little effect on water vapor in the TTL (Dessler et al., 2014).

The water vapor mixing ratio from the trajectory model is gridded into $4° \times 8°$ bins, just as the MLS data were. In the vertical, the trajectory output is binned by averaging the parcels in a pressure range around each MLS or GEOSCCM level. When comparing to MLS, the gridded water vapor mixing ratio is then re-averaged using the MLS averaging kernels following the instruction from Livesey et al. (2017). When doing this kernel averaging, grid boxes with no trajectory parcels (mostly at low altitudes) are filled with monthly water vapor mixing ratios from the reanalyses (ERAi and MERRA-2). Sensitivity tests

confirm that changing water vapor mixing ratio from the reananlyses has no impact on the spatial distribution of the anomalies of TTL water vapor that are the focus of this paper.

## 2.4  Convection clouds

We also use estimates of convective cloud occurrence produced by combining geostationary infrared satellite imagery and microwave rainfall measurements (Pfister et al., 2001; Bergman et al., 2012; Ueyama et al., 2014, 2015). The data have a

15 horizontal resolution of $0.25° \times 0.25°$, a temporal resolution of 3 hours, and cover the period from 2005 to 2016. In this paper, we use the cloud-top height and cloud-top potential temperature to estimate the convective cloud occurrence frequency in the TTL, which we take to be an indicator of convective influence on the TTL. These data are available from https://bocachica.arc. nasa.gov/~lpfister/cloudtop/.

## 3  Results

### 3.1  Influences of the BDC and QBO on TTL water vapor

Fig. 1 shows monthly and tropical averaged 100-hPa water vapor anomalies from MLS observations and from trajectory model runs driven by meteorology from ERAi (traj_ERAi) and MERRA-2 (traj_MERRA2). Similar to the results in Dessler et al. (2013, 2014) at 82 hPa, there is good agreement between the observations and trajectory models at 100 hPa.

  Dessler et al. (2013, 2014) also showed that we can fit tropical average anomalies of 82 hPa water vapor with a simple linear

model:

$$H_2O = a \cdot BDC + b \cdot QBO + c \cdot \Delta T + r, \tag{1}$$

where BDC, QBO, and $\Delta T$ are indices representing the strength of the Brewer-Dobson circulation, the phase of the QBO, and the tropospheric temperature anomalies of the tropical climate system, respectively. Smalley et al. (2017) verified that this approach is also valid in chemistry-climate models.

To gain additional physical insight into the regression result, in this paper we perform a similar multivariable regression, but at individual grid points in the TTL:

$$H_2O(x_i, y_j) = a(x_i, y_j) \cdot BDC + b(x_i, y_j) \cdot QBO + c(x_i, y_j) \cdot \Delta T + r(x_i, y_j). \tag{2}$$

Here, $H_2O(x_i, y_j)$ represents the $H_2O$ anomaly time series at 100 hPa in a grid-box centered at longitude $x_i$ and latitude $y_j$.

The coefficients $a$, $b$, and $c$, as well as the residual term $r$, are also functions of latitude and longitude.

The regressors in Eq. 2 are the same tropical average time series used in Dessler et al. (2013, 2014): BDC is a Brewer-Dobson circulation index — here we use the tropical averaged diabatic heating rate anomaly at 82 hPa, with units of K day$^{-1}$. QBO is a quasi-biennial oscillation index and here we use the standardized monthly and zonally averaged equatorial zonal wind anomaly at 50 hPa, with units of m s$^{-1}$. $\Delta T$ is the tropical averaged tropospheric temperature anomaly at 500 hPa, with

units of degrees K. Because these regressors are tropical average values, they do not vary with location. The QBO index is lagged by 2 months in the regression because the phase of the QBO takes time to impact the TTL temperature and then the water vapor at 100-hPa (Dessler et al., 2013). There is no lag for the BDC and $\Delta T$ indices in this study.

We first analyze MLS water vapor observations. We run the regression on these observations twice: once using BDC and $\Delta T$ regressors from the ERAi reanalysis and again using regressors from the MERRA-2 reanalysis. The QBO index is the

same in both regressions (we use observations downloaded from http://www.cpc.ncep.noaa.gov/data/indices/qbo.u50.index).

The BDC coefficients (Figs. 2a and 2d) are negative over the tropics, consistent with the idea that an enhanced Brewer-Dobson Circulation cools the TTL (Yulaeva et al., 1994) and reduces water vapor (Randel et al., 2006; Dhomse et al., 2008). The QBO coefficients (Figs. 3a and 3d) are positive over almost all of the tropics, as the positive phase of QBO tends to decrease the upwelling in the TTL, thereby warming it (Plumb and Bell, 1982; Davis et al., 2013).

We also run the regression on water vapor simulated by the trajectory model. We use BDC and $\Delta T$ regressors from the same reanalysis used to drive each trajectory model (i.e., we use ERAi fields to regress the ERAi-driven trajectory model); the QBO index is always from the NCEP observations.

The BDC coefficients from regression of the trajectory models (Figs. 2b and 2e) agree well with the coefficients from the regressions of the MLS observations (Figs. 2a and 2d). The average BDC coefficient in the MLS/ERAi regression is -2.6

25    ppmv (K day$^{-1}$)$^{-1}$ (Fig. 2a), in good agreement with the average value from the accompanying trajectory regression, -2.4 ppmv (K day$^{-1}$)$^{-1}$ (Fig. 2b). The gridpoint-by-gridpoint scatter plot (Figs. 2c) demonstrates this agreement in more detail.

The average BDC coefficient in the MLS/MERRA-2 regression is -2.3 ppmv (K day$^{-1}$)$^{-1}$ (Fig. 2d). The average coefficient from the accompanying trajectory regression is -1.4 ppmv (K day$^{-1}$)$^{-1}$ (Fig. 2e). This larger difference stems from what appears to be problems in the MERRA-2 heating rates. These heating rates disagree significantly with those from both ERAi

as well as the original MERRA. Thus, we put more weight on the ERAi results for this coefficient and conclude that the BDC response is well simulated by the trajectory model.

The QBO coefficients from the regressions of the trajectory models are shown in Figs. 3b and 3e; gridpoint-by-gridpoint scatter plots are shown in Figs. 3c and 3f. For the MLS/ERAi comparison (Fig. 3c), the average QBO coefficients are 0.084 ppmv (m s$^{-1}$)$^{-1}$ for the MLS and 0.075 ppmv (m s$^{-1}$)$^{-1}$ for the trajectory model; for the MLS/MERRA2 comparison (Fig.

3f), the average coefficients are $0.14 \ \mathrm{ppmv} \ (\mathrm{m \ s^{-1}})^{-1}$ for the MLS and $0.15 \ \mathrm{ppmv} \ (\mathrm{m \ s^{-1}})^{-1}$ for the trajectory model. As with the BDC comparison, we conclude that the trajectory model does a good job reproducing the regressions of the MLS data.

Overall, the trajectory model accurately captures the impact of the BDC and QBO on TTL water vapor — for both the tropical average and the spatial distribution. This supports the hypothesis that these processes mainly influence TTL water vapor by varying large-scale TTL temperatures and transport (Giorgetta and Bengtsson, 1999; Randel et al., 2000; Geller et al., 2002; Randel et al., 2006; Dhomse et al., 2008; Liang et al., 2011; Davis et al., 2013; Dessler et al., 2013, 2014; Wang et al., 2015), which we expect the trajectory model to reproduce. For this reason, we will not focus any further on these coefficients.

## 3.2 Influence of tropospheric temperature on TTL water vapor

Coefficients of $\Delta T$ from the MLS regressions are mostly positive, with large increases over the Tropical Warm Pool region (TWP) and Indian Ocean (Figs. 4a and 4d), indicating that warming of the tropical troposphere increases TTL water vapor mixing ratio there. Over the Central Equatorial Pacific (CEP), however, a warming troposphere *decreases* TTL water vapor.

The decrease in TTL water vapor in the CEP is not entirely unexpected. TTL temperatures are usually coldest — and water vapor a minimum — above the convection maximum in the TWP. As $\Delta T$ increases in response to an El Niño event, this convective maximum, and its associated TTL cold pool, shifts eastward from the TWP to the CEP (corresponding to the shift from Figs. 5a to 5b) (Davis et al., 2013; Hu et al., 2016; Konopka et al., 2016; Avery et al., 2017). Changes in TTL water vapor are expected to mirror this, with increases in water vapor in the TWP and decreases in the CEP as $\Delta T$ increases.

Both the MLS and trajectory-model regressions show this dipole pattern. However, the MLS regressions yield $\Delta T$ coefficients that are systematically higher than those found in the trajectory-model regressions throughout the tropics (Figs. 4c and 4f). For the MLS/ERAi comparison (Fig. 4c), the average coefficients are $0.43 \ \mathrm{ppmv \ K^{-1}}$ for the MLS and $0.28 \ \mathrm{ppmv \ K^{-1}}$ for the trajectory model; for the MLS/MERRA-2 comparison (Fig. 4f), the average coefficients are $0.20 \ \mathrm{ppmv \ K^{-1}}$ for the MLS and $0.05 \ \mathrm{ppmv \ K^{-1}}$ for the trajectory model. We have also done regressions using tropical average values (using Eq. 1, similar to what was done in Dessler et al. (2013, 2014)) and find that the $\Delta T$ coefficients from MLS and trajectory models are statistically different with probabilities of 85% and 70% for ERAi and MERRA-2, respectively.

We hypothesize that the evaporation of convective ice accounts for the difference between the $\Delta T$ coefficients in the MLS and trajectory-model regressions. As convection moves eastward during an El Niño event, there is an accompanying increase in convective ice in the TTL (as seen in Figs. 5a and 5b; see also Avery et al., 2017), where it evaporates and hydrates the TTL. The moistening from evaporation spreads throughout the tropics and increases the water vapor everywhere. The trajectory model, which does not include this process, simulates a smaller increase in water vapor, leading to smaller $\Delta T$ coefficients (Figs. 4c and 4f).

In support of this, in Fig. 6a we show that the tropical average relative convective cloud occurrence frequency anomalies in the lower stratosphere increase with $\Delta T$. This is consistent with the hypothesis that, as $\Delta T$ increases, we should also see an increase in evaporation of cloud ice.

## 3.3 Tests with a chemistry-climate model

To gain additional confidence in our hypothesis that evaporation of convective ice plays a role in the TTL water budget, we perform a parallel analysis with the GEOSCCM, a model where evaporation of convective ice is known to add water to the TTL (Dessler et al., 2016). To do this, we run a regression on the GEOSCCM 100-hPa water vapor fields as well as on water vapor simulated by a trajectory model driven by the GEOSCCM meteorology.

Figs. 2g and 2h show the spatial distributions of the BDC coefficients from the GEOSCCM and the corresponding trajectory model. This comparison is analogous to the comparison of the regressions on the MLS data and trajectory models driven by reanalyses. The coefficients are similar to each other and to the MLS regressions, suggesting that the GEOSCCM is accurately simulating the impact of BDC changes on TTL water vapor. The influence of the QBO in the version of the GEOSCCM analyzed here does not extend into the TTL, so we have not included a QBO term in the regression and there are consequently no GEOSCCM QBO coefficients in Fig. 3.

Before we discuss the $\Delta T$ coefficients, it is worth pointing out that the GEOSCCM has realistic ENSO variability in TTL temperatures and convective ice. Figs. 5c and 5d show that the monthly convective cloud ice water content (IWC) anomalies at 118 hPa and cold anomalies at 100 hPa in the GEOSCCM shift eastward as $\Delta T$ warms from a cold phase (Fig. 5c) to a warm phase (Fig. 5d), just as they did in observations (Figs. 5a and 5b).

The $\Delta T$ coefficient fields from the GEOSCCM and associated trajectory-model regressions (Figs. 4g and 4h) show the same structural differences as do the $\Delta T$ coefficients from the MLS and accompanying trajectory-model regressions — that the $\Delta T$ coefficient is larger in the GEOSCCM regression than in the trajectory-model regression — as in the observations, the tropical average $\Delta T$ coefficients from GEOSCCM and trajectory model are significantly different at the 85% confidence level.

In the last section, we hypothesized that this difference in the coefficients was due to evaporation of convective ice in the MLS data, a process not included in the trajectory model. To directly test this hypothesis in the GEOSCCM, we run a second version of the trajectory model that includes the evaporation of convective ice from GEOSCCM, referred to hereafter as traj_ccm_ice. To do this, we use the 6-hourly three-dimensional convective cloud IWC field from GEOSCCM and linearly interpolate it to each parcel's position at every time step. We then assume instantaneous and complete evaporation of this ice into the parcel by adding the IWC to the parcel's water vapor, although we do not let parcels' water vapor exceed 100% relative humidity with respect to ice. This is the same procedure used to simulate convective ice evaporation by Dessler et al. (2016) .

We then run the regression on the traj_ccm_ice's water vapor field. The scatter plot of GEOSCCM vs. traj_ccm_ice BDC coefficients (Fig. 2k) shows larger scatter than the comparison without ice (Fig. 2i). The increase in scatter is likely the result of the crudeness of our microphysical assumptions, particularly the assumption that convective ice evaporates instantaneously. However, the comparison between the tropical average GEOSCCM BDC coefficient, -6.2 ppmv $(\mathrm{K\,day^{-1}})^{-1}$, and those from the trajectory models, -5.8 and -6.9 ppmv $(\mathrm{K\,day^{-1}})^{-1}$ without and with convective ice evaporation, respectively, is similar.

The scatter plot of GEOSCCM vs. traj_ccm_ice $\Delta T$ coefficients (Fig. 4k) similarly shows larger scatter than the comparison without ice (Fig. 4i). Adding ice does, however, increase the average $\Delta T$ coefficient (seen by comparing Figs 4i to 4k), from 0.16 ppmv $\mathrm{K^{-1}}$ to 0.32 ppmv $\mathrm{K^{-1}}$, bringing the trajectory model into closer agreement with the GEOSCCM, which has a

corresponding value of $0.31$ ppmv $K^{-1}$. There are also some interesting changes in the spatial pattern of the traj_ccm_ice (Fig. 4j). For example, negative $\Delta T$ coefficients appear in the TWP and Indonesia in the traj_ccm_ice regression; The cause of this is unknown, but also is likely linked to the trajectory model's ice evaporation assumption.

We showed in the previous section that the convective cloud occurrence frequency in the TTL increased as $\Delta T$ increased (Fig. 6a) and we also see that the GEOSCCM simulates a similar correlation between convective cloud IWC and $\Delta T$ (Fig. 6b). While these are not exactly the same quantity, they show a consistency that provides confidence that the behavior of the model is realistic.

Finally, to quantify convective ice evaporation, we calculate the evaporation rate of convective ice at 100 hPa in the trajectory model. To do this, we save the amount of water added to each parcel by ice evaporation in every time step. We then bin and average the amount evaporated to come up with the distribution of the amount evaporated per day. Note that much of this water added will be lost in subsequent dehydration events, so this does not represent net water added to the stratosphere.

Fig. 6c shows that the tropical average evaporation rate of convective ice also increases with $\Delta T$, which provides further evidence that the difference in $\Delta T$ coefficients between the GEOSCCM and the trajectory model is due to evaporation of convective ice. Fig. 7 shows the distribution of monthly averaged evaporation rate during ENSO-like cold and warm phases in the GEOSCCM. We see that, as $\Delta T$ increases and we transition from a cold to warm phase of variability, the location of ice evaporation shifts from the TWP and Indian Ocean to the CEP. This is consistent with the analysis of Avery et al. (2017).

## 4   Conclusions

Previous work has shown that TTL water vapor variability is mainly controlled by TTL temperature variability (Mote et al., 1996; Holton and Gettelman, 2001; Fueglistaler et al., 2009). In particular, variations in the Brewer-Dobson circulation (BDC) and the quasi-biennial oscillation (QBO) play key roles (Fueglistaler and Haynes, 2005; Geller et al., 2002; Liang et al., 2011; Liess and Geller, 2012; Calvo et al., 2010; Randel et al., 2006; Dessler et al., 2013, 2014; Tao et al., 2015). It has also been suggested by many previous investigators that evaporation of convective ice may contribute water vapor to the TTL (Khaykin et al., 2009; Hassim and Lane, 2010; Carminati et al., 2014; Frey et al., 2015; Virts and Houze, 2015; Schoeberl and Dessler, 2011; Schoeberl et al., 2014; Ueyama et al., 2014, 2015; Dessler et al., 2016; Avery et al., 2017). In this paper, we analyze the spatial distribution of TTL water vapor and conclude that, indeed, convective ice evaporation makes a small contribution to the interannual variability over the MLS period.

To do that, we use a linear regression model on TTL water vapor at individual grid points over the tropics to decompose the spatial distribution of TTL water vapor variability into contributions from changes in the BDC, QBO, and tropospheric temperature ($\Delta T$). We run this linear regression model on MLS observations of TTL water vapor anomalies and on water vapor anomalies simulated by a trajectory model that only includes the effects of TTL temperatures on water vapor.

The spatial patterns and magnitudes of the BDC and QBO coefficients agree well between MLS observations and associated trajectory model simulations. This confirms that these processes affect TTL water vapor mainly by changing TTL temperatures

(Randel et al., 2000; Geller et al., 2002; Randel et al., 2006; Dhomse et al., 2008; Liang et al., 2011; Davis et al., 2013; Dessler et al., 2013, 2014).

The spatial distribution of $\Delta T$ coefficients has an obvious dipole structure associated with the ENSO (Konopka et al., 2016): negative values in the Central Equatorial Pacific (CEP), where temperatures decrease as the troposphere warms, and positive values in the Tropical Warm Pool (TWP), where the opposite occurs.

We also find that $\Delta T$ coefficients from the MLS observations are larger throughout the tropics than in the trajectory model simulations. We hypothesize that increases in convection as $\Delta T$ increases lead to increases in evaporation of convective ice in the TTL. This increases the $\Delta T$ coefficient in the MLS analysis, but not in the trajectory model, which does not have convective ice evaporation in it. We see support for this in the observations of increased convective cloud occurrence frequency in the TTL as $\Delta T$ increases. This result is also in agreement with the case study in Avery et al. (2017) as well as the model analysis in Schoeberl and Dessler (2011), Schoeberl et al. (2014), and Ueyama et al. (2014, 2015).

To gain additional confidence in our hypothesis that evaporation of convective ice is responsible for the difference in $\Delta T$ coefficients, we test the methodology in a parallel analysis with the GEOSCCM, a chemistry-climate model where evaporation of convective ice is known to add water to the TTL (Dessler et al., 2016). We find that the results of this analysis show the same difference — that the $\Delta T$ coefficients from the regression of the GEOSCCM's water vapor field are larger than the coefficients from a trajectory model driven by GEOSCCM meteorology.

We confirm this is due to evaporation of convective ice by running a second version of the trajectory model, which includes convective ice evaporation. We find that the $\Delta T$ coefficient from the regression of this version of the trajectory model is in agreement with that from the GEOSCCM regression.

Putting all of these together, we conclude that variability in the evaporation of convective ice plays a role in water vapor variability in the TTL. Our work should not be taken as opposing previous research (Randel et al., 2006; Schiller et al., 2009; Wright et al., 2011; Randel and Jensen, 2013) that concluded that most of the variance in TTL water vapor over the last few decades is due to TTL temperatures. We concur that the impact of convective ice only is a minor contributor to TTL water vapor variability over the period spanned by the MLS data. But for GEOSCCM, which does an excellent job simulating TTL water vapor over the comparable period, suggests that convective ice may play a much larger role in long-term trends of TTL and stratospheric water vapor (Dessler et al., 2016), so more research on this phenomenon is clearly warranted.

*Competing interests.* The authors declare that they have no conflict of interests.

*Acknowledgements.* We thank Dr. Mark Schoeberl for his insights into this problem. This work was supported by NASA grant NNX14AF15G to Texas A&M University. We would like to thank Dr. Luke Oman and Dr. Anne Douglass for providing the simulation of GEOSCCM used in this study, the modeling effort is supported by the NASA MAP program and the high-performance computing resources that were provided by the NASA Center for Climate Simulation (NCCS).

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

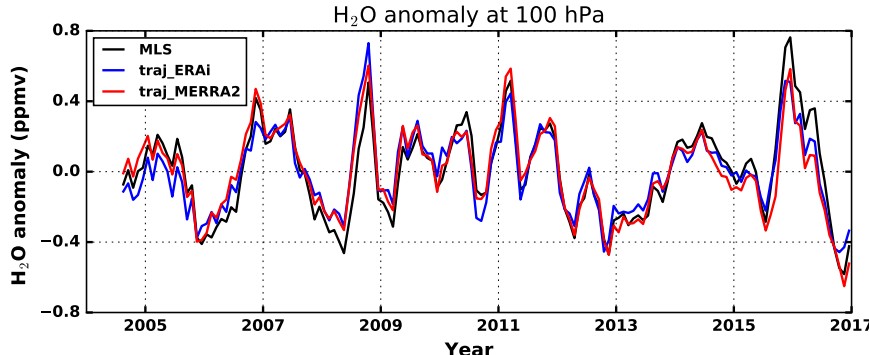

**Figure 1.** Tropical average (30°N-30°S) monthly water vapor anomalies at 100 hPa from MLS observations (black line) and from trajectory model runs driven by ERAi (blue line) and MERRA-2 (red line) from August 2004 through 2016. Anomalies are calculated by subtracting the mean annual cycle.

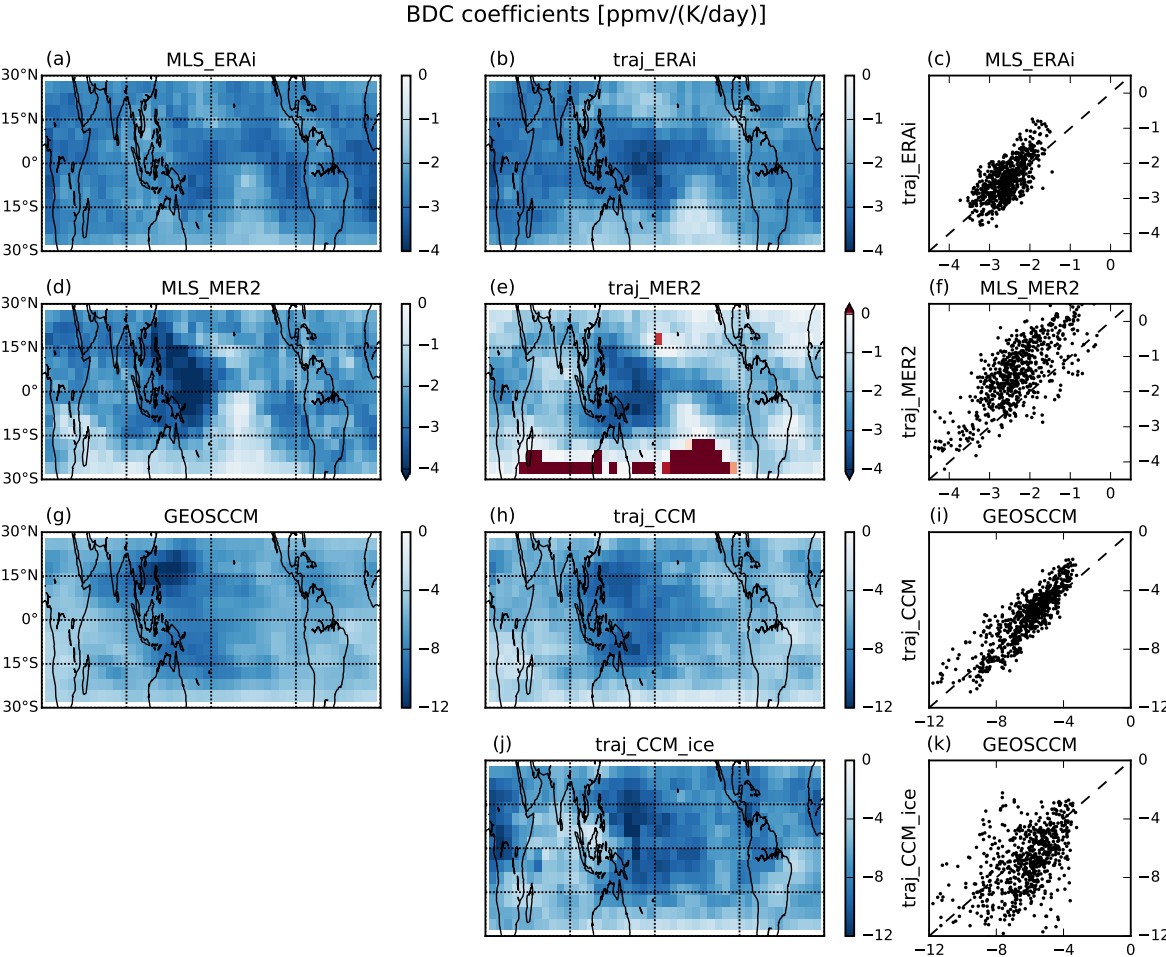

**Figure 2.** Coefficients of the BDC regressor from MLS and GEOSCCM water vapor fields (left column), as well as the coefficients from regression of the associated trajectory model fields (middle column). Scatter plots of MLS/GEOSCCM regressions vs. trajectory model regressions indicate the similarity of the fields (right column). The MLS and associated trajectory-model regressions cover the period August 2004 to December 2016 between 30°N and 30°S. The GEOSCCM and associated trajectory-model regressions cover 2005-2016 model years. The bottom row shows coefficients from regressions of a run of the trajectory model driven by GEOSCCM meteorology that includes evaporation of convective ice.

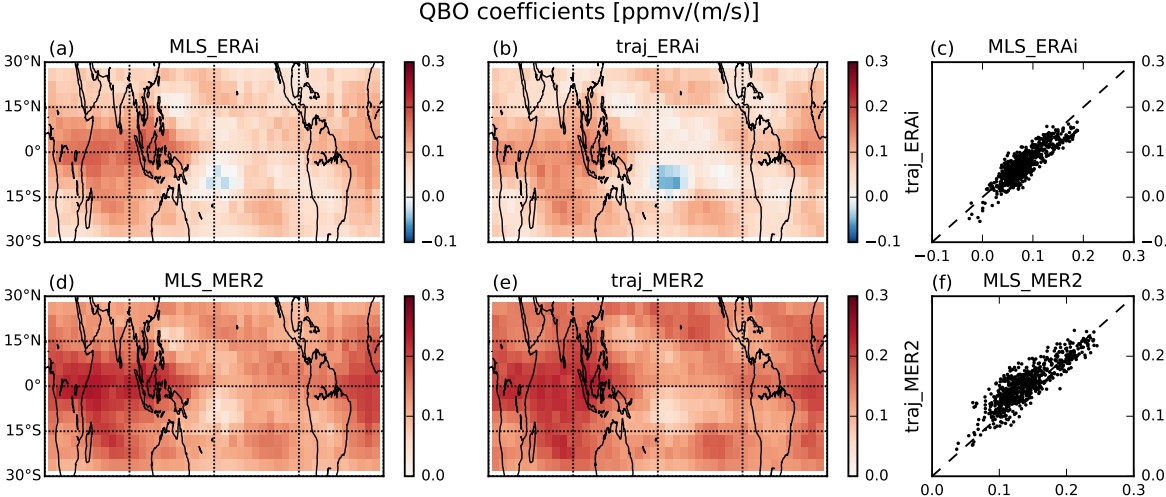

**Figure 3.** Same as Fig. 2, but for coefficients of the QBO regressor.

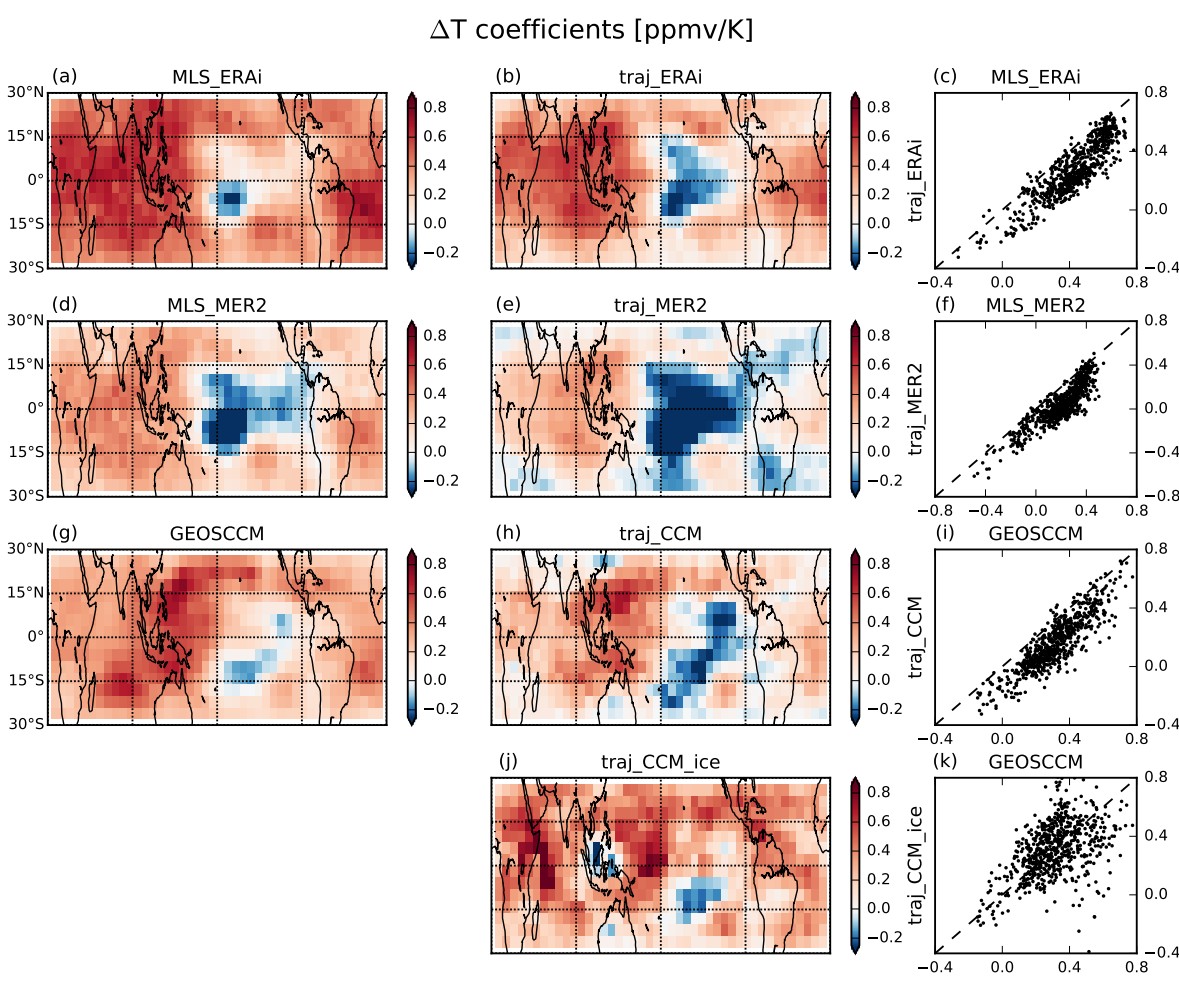

**Figure 4.** Same as Fig. 2, but for the coefficient of the $\Delta T$ regressor.

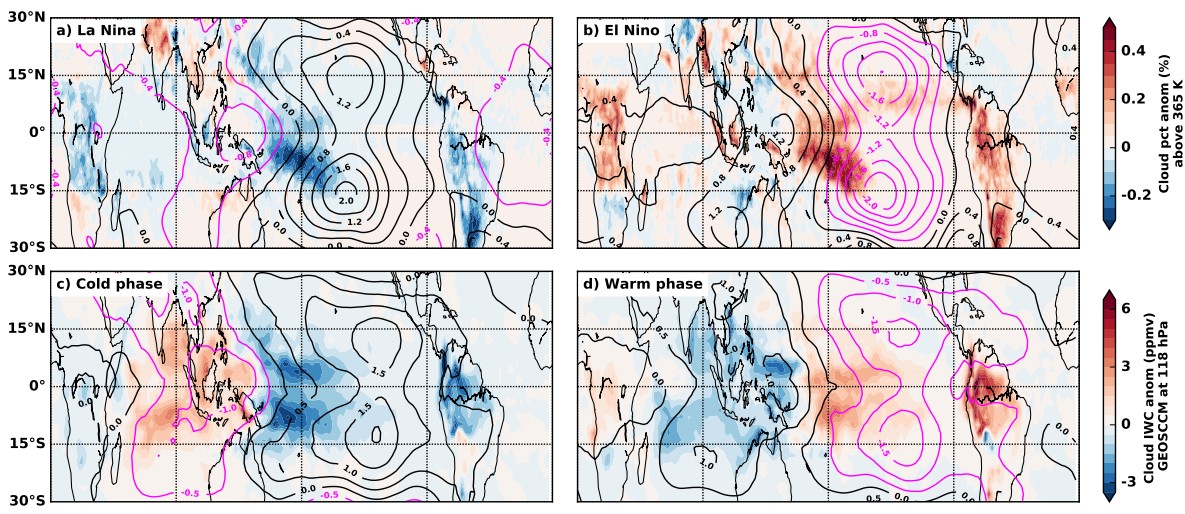

**Figure 5.** Averaged monthly cloud occurrence frequency anomalies above 365 K during (a) La Niña and (b) El Niño months from 2005 to 2016; also shown as contours are temperature anomalies at 100 hPa. La Niña and El Niño months are based on the NOAA Oceanic Niño Index (ONI) in the Niño 3.4 region (5°S to 5°N; 170°W to 120°W). Averaged monthly GEOSCCM convective cloud ice water content (IWC) anomalies (ppmv) at 118 hPa during (c) cold and (d) warm GEOSCCM phases from model years 2005 to 2016 with averaged temperature anomalies at 100 hPa shown as contours. The cold and warm phases are defined to be GEOSCCM surface temperature anomaly (departures from the mean annual cycle) of at least -0.5 K and +0.5 K, respectively, in the Niño 3.4 region (same as ONI).

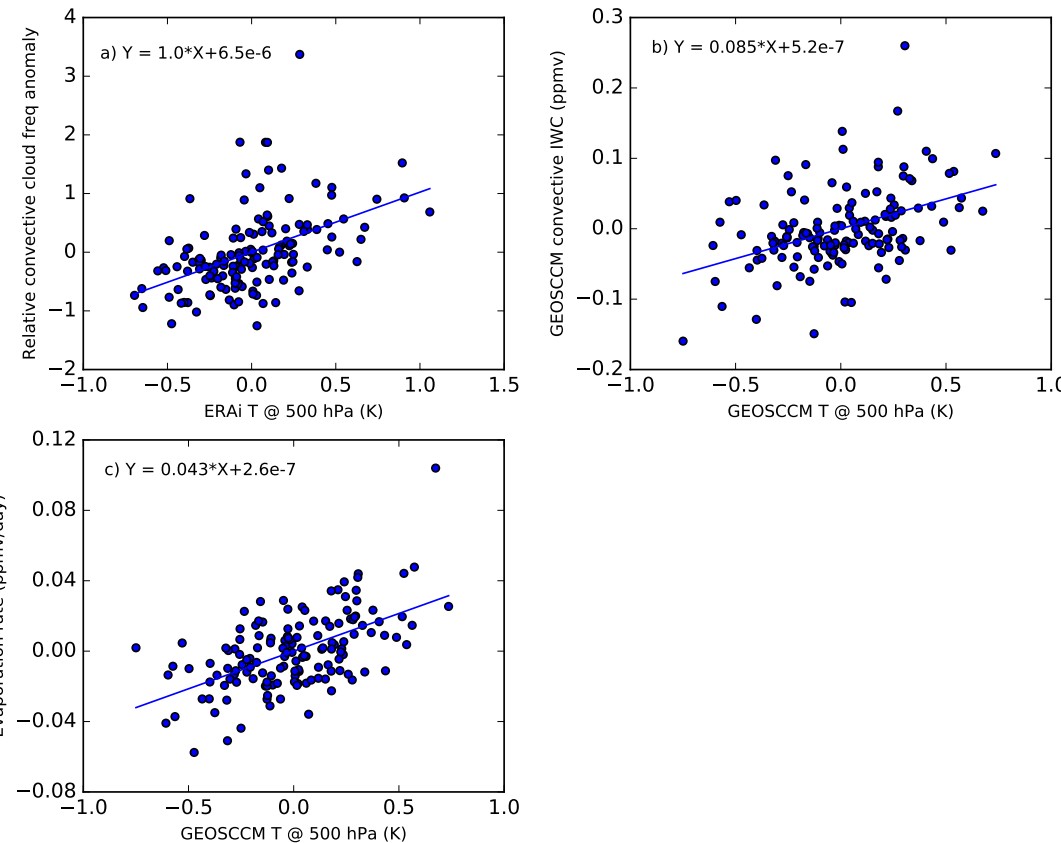

**Figure 6.** (a) Scatter plot of observed convective cloud occurrence frequency anomalies at 390 K vs. 500-hPa $\Delta T$ from ERAi. Cloud frequency is expressed as the percent change relative to the average cloud frequency at this level ($1.2 \times 10^{-6}$ in tropics). (b) Scatter plot of convective IWC anomalies at 100 hPa vs. $\Delta T$ from GEOSCCM. (c) Scatter plot of GEOSCCM convective cloud evaporation rate anomalies at 100 hPa vs. $\Delta T$. All the data are monthly and tropical averaged from 30°S to 30°N. The straight lines are least-squares fits to the data.

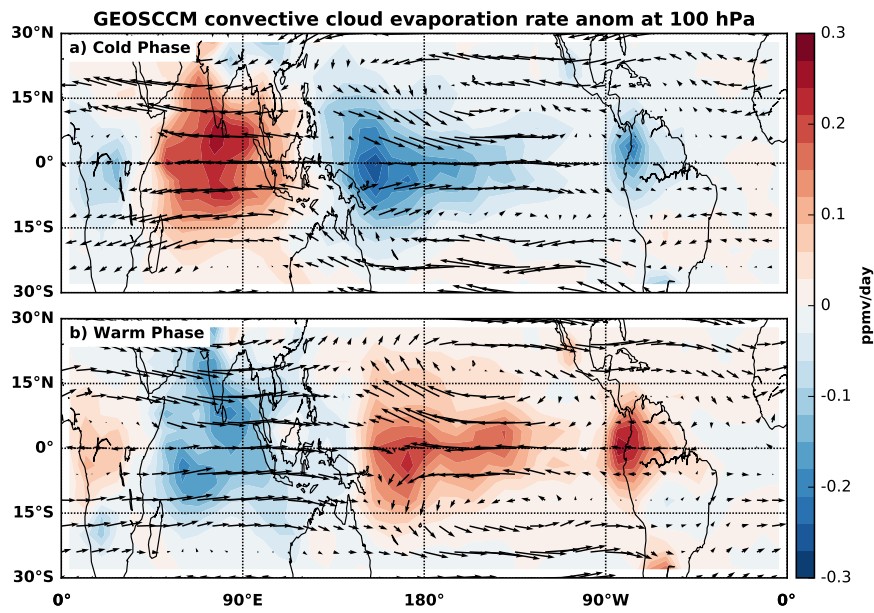

**Figure 7.** Averaged monthly GEOSCCM convective ice evaporation rate anomalies at 100 hPa during (a) cold and (b) warm GEOSCCM phases from 2005 to 2016. Also shown are averaged horizontal wind anomaly vectors at 100 hPa.