# Peer review of "Effects of convective ice evaporation on interannual variability of tropical tropopause layer water vapor"

_Atmospheric Chemistry and Physics, 2017_

## Referee Comment (RC1) · Anonymous Referee #1 · 16 Nov 2017

Summary:

This paper expands on the regression techniques employed in previous work by one of the coauthors to explore spatial variations in the tropics. The paper seeks to draw some conclusions about the importance of convection in explaining variations in TTL water based on comparing the regressions of water measurements and water calculated from a trajectory model. Comparison with climate model water vapor simulations are also included.

General Comment:

Though the trajectory model physics is greatly oversimplified and clearly NOT state-of-the art, the approach has the potential to be useful. Unfortunately, the authors focus

on only one aspect of the results of their method, namely that convection moistens the TTL. This is certainly true, but has been previously established by a much more detailed and dynamically and microphysically realistic trajectory model (Ueyama et al, 2015).

The paper also shows that much of the spatial variability in the tropics can be explained by the regression variables (BDC, QBO, and tropospheric temperature). This second result is actually quite interesting, but its implications are not explored. For example, Figure 2 (regression with the BD circulation) shows some consistent spatial patterns in the regression coefficient. This is also true for the QBO coefficients (Figure 3). However, only the spatial variations of the tropospheric temperature coefficients are explored (to make the point about convection moistening the TTL). At the very least, there should be some discussion of these patterns (QBO and BDC coefficient patterns) and what they might mean, consistent with the objectives set out in the last paragraph of the introduction. Arguably the concentration of negative BDC coefficients in the western Pacific may not be a surprise, but why the QBO coefficients are strongest in the Indian ocean is puzzling.

To make this paper acceptable for publication, the authors need to do two things (as well as address the specific points below).

(1) As indicated above, the major result that the authors stress is the moistening of the TTL by convection. The authors need to be mindful (and point out clearly) that this is not new. What may be considered new is: (1) moistening in the central Pacific associated with El Nino (the Avery work is only a case study, so the general point is new); and (2) the method (regression) by which this result is arrived at. The authors need to clearly emphasize what is new and what is not.

(2) Explore the other implications of their regression analysis (see second paragraph above). The authors may want to save this for another paper, but ignoring most of the results of their regression analysis is simply unacceptable.

Specific points:

Page 2, Line 10: How good is Dessler's model in evaluating the increase in water vapor in the UTLS due to convection for climate models? I would argue that we really can't simulate convection to the appropriate level of detail to get the effect of convective injection on UTLS water vapor in the current climate, so it is going to be difficult to make forecasts. Dessler' 2016 paper is one of the motivators to looking at this problem, but the statement is too strong. "a significant fraction of this increase may be due to the evaporation of lofted ice..." is more appropriate.

Page 7, lines 12-15: Look at the figure! The BDC coefficients change quite a bit over continental areas, so the statement is wrong. The statement that this is due to instant evaporation needs to be supported by some reasoning or evidence.

Page 7, lines 15-16: Systematic differences are reduced (comparing 4i to 4k), but scatter is much larger. For the BDC coefficient, the agreement (2i and 2k) is actually substantially worse. A region of negative coefficients appears over Indonesia in Figure 4j. I guess that is not statistically significant, but neither are any of the negative regions in figures 4g, 4j, and 4h that are emphasized in showing how convection moistens the central Pacific. There needs to be more discussion of these points.

Page 7, last paragraph: The contents of this paragraph should be moved to the conclusions section. The point about long-term trends of stratospheric water vapor (which this paper does not address at all) is speculation. Speculation is OK, but not in the results section.

Page 7, line 33: How is it shown that a free running climate model GEOSCCM simulates TTL water vapor "over this period?" Since it is NOT a reanalysis, this needs to be explained.

Page 7, Line 4: I would use a study with a more detailed realistic model (e.g., Ueyama et al, 2015) to make this point.

[Figure]

Minor comments:

line 7: ...as THE troposphere warms...

Figure 3: Axes are mislabeled in (c) and (f)

Figure 5 caption: magenta is negative, and black is positive temperature anomalies (reverse of what is in the caption).

The black dots of significance are not always easy to see in the figures, suggest another color.
* * *

---

## Referee Comment (RC2) · Anonymous Referee #2 · 30 Nov 2017

This paper explores variability of TTL water vapor from satellite observations, trajectory model calculations and GEOSCCM results, aiming to make a convincing argument for a direct effect of deep tropical convection. The calculations are based on multiple regression analysis of the different data sets, and the key points regarding convection are deduced from the regression coefficients related to tropospheric temperature (500 hPa zonal average temperature). The trajectory model does an excellent overall job of simulating interannual changes in water vapor, which is due to the accurate tropical tropopause temperatures input to the model. However, there are small systematic differences between the regression of the observed MLS water vapor onto tropospheric temperature and the results from the trajectory model, and the authors interpret this difference as evidence for the impact of deep convection. The results are

repeated using GEOSCCM model simulations with similar results, and further analyses of the GEOSCCM model with parameterized convective ice are used to estimate the ice contribution to stratospheric water vapor. While the different results point to possible effects related to deep convection, a convincing argument is not made based on the tropospheric temperature regressions (which I believe are mainly reflecting ENSO variations, as discussed below). In fact, there is no clear discussion of why the effects of deep convection should show up primarily in the tropospheric temperature regressions, as opposed to the other regression terms. My recommendation is that the authors provide more convincing evidence before the paper is acceptable for publication.

Specific comments:

The regression model, based on BDC, QBO and dT parameters, is an extension of Dessler et al 2014 (D14). The accurate simulation of H2O in the trajectory model is evidence that tropopause temperatures primarily control H2O (as acknowledged here), and the regression model then accounts for variability of tropopause temperature. This is why the BDC accounts for most of the H2O variance, as the BDC (heating rates) are closely proportional to temperature. The component of H2O variance tied to tropospheric temperatures (dT) is relatively small in the regression model, with larger relative uncertainties (the corresponding H2O variations for dT in Fig. 4 are < 0.1 ppmv, versus $\sim$ 0.5 ppmv for the BDC in Fig. 2). Time series of dT (Fig. 4 in D14) show that dT is mainly a proxy for ENSO variability, which explains the see-saw spatial structures in Fig. 4 (consistent with the patterns in Figs. 5-6). This ENSO spatial structure was discussed recently in Konopka et al, 2016, JGR, which should be referenced.

The key points of this paper relate to the small differences between the dT regression fits for MLS observations (or GEOSCCM model) and trajectory model results. To be convincing, the authors need to explain why the convection effect (persistent moistening) is associated with the dT (ENSO) regression, and demonstrate links to observed convection. Is there in fact more convection (in a global sense) when the troposphere is warm? The dT regression differences (e.g. Fig. 4a vs. 4b) are likely within the uncertainty estimates of the regression fits, although this is not discussed. Furthermore, scatter plots (Figs. 4c,f,i) suggest an overall shift of the coefficients that is not dependent on location, and in particular the differences are not evidently related to regions of deep convection. Given these uncertainties, the argument that the differences are due to the neglected effects of deep convection are unconvincing.

My suggestion for revising the paper: 1) The authors could keep the present analysis, but provide more convincing discussion regarding the physical relationship between convection and dT, and in addition demonstrate statistical significance of the dT regression differences, and show clear physical links to observed convection. 2) A more convincing argument could be made by systematically analyzing the differences between observations and trajectory model results, and demonstrating that these differences are consistent with convective influence (e.g. using their spatial and temporal characteristics, and links with observed convection).
* * *

---

## Author Comment (AC1) · 11 Jan 2018

**General Comments:**

To make this paper acceptable for publication, the authors need to do two things (as well as address the specific points below).

(1) As indicated above, the major result that the authors stress is the moistening of the TTL by convection. The authors need to be mindful (and point out clearly) that this is not new. What may be considered new is: (a) moistening in the central Pacific associated with El Nino (the Avery work is only a case study, so the general point is new); and (b) the method (regression) by which this result is arrived at. The authors need to clearly emphasize what is new and what is not.

[Figure]

Response: We have revised the introduction to clarify how our work fits into the previous literature. See page 2, lines 8-14 and lines 20-27.

(2) Explore the other implications of their regression analysis (see second paragraph above). The authors may want to save this for another paper, but ignoring most of the results of their regression analysis is simply unacceptable.

Response: In our opinion, the spatial patterns of the QBO and BDC coefficients are not interesting enough to warrant additional discussion. Thus, we have not added anything to the paper in response to this comment. We are open, however, to suggestions from the reviewer about what other implications are noteworthy.

**Specific points:**

Page 2, Line 10: How good is Dessler's model in evaluating the increase in water vapor in the UTLS due to convection for climate models? I would argue that we really can't simulate convection to the appropriate level of detail to get the effect of convective injection on UTLS water vapor in the current climate, so it is going to be difficult to make forecasts. Dessler's 2016 paper is one of the motivators to looking at this problem, but the statement is too strong. "a significant fraction of this increase may be due to the evaporation of lofted ice..." is more appropriate.

Response: This is covered in some detail in Dessler et al. (2016). Our assessment is that that analysis was able to unambiguously identify the influence of convective ice evaporation in the models. That said, we now note that this analysis only analyzed two models, so we've modified some text to make this clearer: "a significant fraction of this increase was found to be due to the evaporation of convective ice from convection in two chemistry-climate models (Dessler et al., 2016)" (Page 2, lines 16-18).

Page 7, lines 12-15: Look at the figure! The BDC coefficients change quite a bit over

continental areas, so the statement is wrong. The statement that this is due to instant evaporation needs to be supported by some reasoning or evidence.

Response: We agree this was inartfully worded. We changed the statement to "The scatter plot of GEOSCCM vs. traj_ccm_ice BDC coefficients (Fig. 2k) shows larger scatter than the comparison without ice (Fig. 2i). The increase in scatter is likely the result of the crudeness of our microphysical assumptions, particularly the assumption that convective ice evaporates instantaneously. However, the comparison between the tropical average GEOSCCM BDC coefficient, -6.2 ppmv $(\mathrm{K\,day}^{-1})^{-1}$, and those from the trajectory models, -5.8 and -6.9 ppmv $(\mathrm{K\,day}^{-1})^{-1}$ without and with convective ice evaporation, respectively, is similar." (page 7, lines 23-28).

Page 7, lines 15-16: Systematic differences are reduced (comparing 4i to 4k), but scatter is much larger. For the BDC coefficient, the agreement (2i and 2k) is actually substantially worse. A region of negative coefficients appears over Indonesia in Figure 4j. I guess that is not statistically significant, but neither are any of the negative regions in figures 4g, 4j, and 4h that are emphasized in showing how convection moistens the central Pacific. There needs to be more discussion of these points.

Response: We have re-written this discussion to highlight that the scatter has visibly increased in the coefficient scatter plots when we add ice (page 7, lines 29-34).

Page 7, last paragraph: The contents of this paragraph should be moved to the conclusions section. The point about long-term trends of stratospheric water vapor (which this paper does not address at all) is speculation. Speculation is OK, but not in the results section.

Response: Done.

Page 7, line 33: How is it shown that a free running climate model GEOSCCM simulates TTL water vapor "over this period"? Since it is NOT a reanalysis, this needs to be explained.

Response: We have modified this statement and added some discussion (See page 3, lines 13-16 and page 9, lines 21-24).

Page 7, Line 4: I would use a study with a more detailed realistic model (e.g., Ueyama et al, 2015) to make this point.

Response: We have re-written this sentence as "In the last section, we hypothesized that this difference in the coefficients were due to evaporation of convective ice in the MLS data, a process not included in the trajectory model" (page 7, lines 14-16).

**Minor comments:**

line 7: ...as THE troposphere warms...

Response: Added.

Figure 3: Axes are mislabeled in (c) and (f)

Response: Corrected.

Figure 5 caption: magenta is negative, and black is positive temperature anomalies (reverse of what is in the caption). The black dots of significance are not always easy to see in the figures, suggest another color.

Response: We have removed the dots from the plot. We did this because what is important in the plots is not whether the coefficients are non-zero, but rather their overall spatial pattern. In its place, we've added a discussion about the significance of

differences in the tropical average quantities (page 6, lines 16-20 and page 7, lines 13-14).

---

## Author Comment (AC2) · 11 Jan 2018

**Specific comments:**

The regression model, based on BDC, QBO and $\Delta T$ parameters, is an extension of Dessler et al., 2014 (D14). The accurate simulation of $H_2O$ in the trajectory model is evidence that tropopause temperatures primarily control $H_2O$ (as acknowledged here), and the regression model then accounts for variability of tropopause temperature. This is why the BDC accounts for most of the $H_2O$ variance, as the BDC (heating rates) are closely proportional to temperature. The component of $H_2O$ variance tied to tropospheric temperatures ($\Delta T$) is relatively small in the regression model, with larger relative uncertainties (the corresponding $H_2O$ variations for $\Delta T$ in Fig. 4 are < 0.1

ppmv, versus $\sim$ 0.5 ppmv for the BDC in Fig. 2). Time series of $\Delta T$ (Fig. 4 in D14) show that $\Delta T$ is mainly a proxy for ENSO variability, which explains the see-saw spatial structures in Fig. 4 (consistent with the patterns in Figs. 5-6). This ENSO spatial structure was discussed recently in Konopka et al., 2016, JGR, which should be referenced.

Response: The reference (Konopka et al., 2016) has been added (page 6, line 11 and page 9, line 1).

The key points of this paper relate to the small differences between the $\Delta T$ regression fits for MLS observations (or GEOSCCM model) and trajectory model results. To be convincing, the authors need to explain why the convection effect (persistent moistening) is associated with the $\Delta T$ (ENSO) regression, and demonstrate links to observed convection. Is there in fact more convection (in a global sense) when the troposphere is warm?

Response: To demonstrate the correlation between convection and the tropical warming, we added a scatter plot of tropical average convective cloud occurrence frequency at 370 K from observation and 500 hPa $\Delta T$ (Fig. 6a); it shows that that the convective cloud occurrence frequency in the TTL increases with $\Delta T$.

This also occurs in the models. Dessler et al. (2016) (their Fig. 2) showed that convective ice in models' TTL increases in response to long-term warming. In this paper, we have also added a plot showing that IWC and net ice evaporation both increase with $\Delta T$ in response to interannual variability (Figs. 6b and 6c).

We also tested the correlation between convection and other regressors, i.e. BDC and QBO, and there is no apparent correlation like what found between convection and the tropical warming.

Overall, we view this as a reasonable assumption.

[Figure]

The $\Delta T$ regression differences (e.g. Fig. 4a vs. 4b) are likely within the uncertainty estimates of the regression fits, although this is not discussed.

Response: We have added a discussion about statistics testing of how confidently we can conclude that the tropical average coefficients are different (page 6, lines 16-20 and page 7, lines 13-14).

Furthermore, scatter plots (Figs. 4c,f,i) suggest an overall shift of the coefficients that is not dependent on location, and in particular the differences are not evidently related to regions of deep convection. Given these uncertainties, the argument that the differences are due to the neglected effects of deep convection are unconvincing.

Response: The reviewer makes a good point. While the hydration due to convection is localized, the impact is indeed spread throughout the tropics. That was not clear in the previous version and we have made changes throughout the paper to better reflect this.

My suggestion for revising the paper: 1) The authors could keep the present analysis, but provide more convincing discussion regarding the physical relationship between convection and $\Delta T$, and in addition demonstrate statistical significance of the $\Delta T$ regression differences, and show clear physical links to observed convection.

Response: As discussed above, we have added new figures to connect $\Delta T$ and convective ice in observations and models (Figs. 6b and 6c). We also show that the ice evaporation rate in the GEOSCCM also increases with $\Delta T$ (Fig. 6c). While somewhat circumstantial, we feel the case we've made is nonetheless convincing.

2) A more convincing argument could be made by systematically analyzing the differences between observations and trajectory model results, and demonstrating that
these differences are consistent with convective influence (e.g. using their spatial and temporal characteristics, and links with observed convection).

Response: This is not a new idea. One of us (AED) tried to do something like this about 10 years ago and it just didn't work. We know that other groups (such as one at JPL) also tried doing this. The main problem is that the TTL is relatively close to saturation, so any individual convective event doesn't add that much water. As a result, it's hard to pull the signal of that out of the background noise, which is considerable due to trajectory uncertainty and noise in the individual MLS measurements. Because of this prior experience, we do not judge this is a profitable course of research.

---

## Author Response (AR2)

**Reply to anonymous Referee #1**

The authors' lack of interest in the spatial variations of the BDC and QBO regressors should not hinder publication (referring to their response to my second recommendation in the initial review). However, the spatial variations in the delta T regression coefficients (Figure 4) are a key part of the paper's argument that convection is moistening the TTL.

The authors need to give a quantitative explanation of why the spatial variations in the BDC and QBO coefficients are less important for water vapor variability than the spatial variations in the delta T coefficient.

Response: As discussed on page 6 lines 3-7, the trajectory model accurately reproduces the features of the BDC and QBO coefficients (see the consistency shown in the scatter plots in Figs. 2 and 3). That is not the case for the T coefficient, which is why we focus on that one. We already make this point, but we?ve added one sentence to make this explicit (page 6 lines 7-8).

We have also added tropical average BDC and QBO coefficients from observations and trajectory model simulations in section 3.1 to emphasize the good quantitative agreement (page 5 lines 24-26, 27-28 and page 5 line 33- page 6 line 1):

Page 5 Lines 24-26: "The average BDC coefficient in the MLS/ERAi regression (Fig. 2a) is -2.6 ppmv $(\text{K day}^{-1})^{-1}$, in good agreement with the average value from the accompanying trajectory regression (Fig. 2b), -2.4 ppmv $(\text{K day}^{-1})^{-1}$"

Page 5 Lines 27-28: "The average BDC coefficient in the MLS/MERRA-2 regression (Fig. 2d) is -2.3 ppmv $(\text{K day}^{-1})^{-1}$. The average coefficient from accompanying trajectory regression (Fig. 2e) is -1.4 ppmv $(\text{K day}^{-1})^{-1}$."

Page 5 Line 33- Page 6 Line 1: "For the MLS/ERAi comparision (Fig. 3c), the average QBO coefficients are 0.084 ppmv $(\text{m s}^{-1})^{-1}$ and 0.075 ppmv $(\text{m s}^{-1})^{-1}$; for the MLS/MERRA2 comparison (Fig. 3f), the average coefficients are 0.14 ppmv $(\text{m s}^{-1})^{-1}$ and 0.15 ppmv $(\text{m s}^{-1})^{-1}$."

**Reply to anonymous Referee #2**

The authors have done a reasonable job of addressing my previous comments and revising the paper. I appreciate the new results in Fig. 6a showing how observed convective cloud occurrence frequency changes with mean tropospheric temperature, but I would like to see a little more detail on these results in the final version of the paper.

In particular, the variations in convective cloud frequency at 370 K in Fig. 6a are $\sim 0.01\%$, and I wonder how large this signal is compared to background values (i.e. how large are the fractional variations?).

Also, since the cold point is above 370 K (e.g. Seidel et al, 2001, JGR), how do these observed variations at 370 K mesh with the statement on p. 2, line 6 that "only convection reaching above the cold point is likely to significantly impact the humidity of the stratosphere"?

Response: To address these two linked comments, we first now plot data at 390 K instead of 370 K. We also changed the GEOSCCM convective IWC from 118 hPa to 100 hPa. The conclusions are not altered by this change.

To better show the magnitude of the changes, we now plot the percent change in cloud frequency in Fig. 6a (the convective cloud frequency anomaly divided by the mean convective cloud frequency).

We have also added the magnitude of the tropical average convective cloud occurrence frequency at 390 K to the text (Page 20).